# Role of Post-Translational Modifications of cGAS in Innate Immunity

**DOI:** 10.3390/ijms21217842

**Published:** 2020-10-22

**Authors:** Yakun Wu, Shitao Li

**Affiliations:** Department of Microbiology and Immunology, Tulane University, New Orleans, LA 70118, USA; ywu36@tulane.edu

**Keywords:** cGAS, innate immunity, post-translational modification, phosphorylation, ubiquitination, acetylation, glutamylation, sumoylation

## Abstract

Cyclic GMP–AMP synthase (cGAS) is the synthase that generates the second messenger cyclic GMP–AMP (cGAMP) upon DNA binding. cGAS was first discovered as the cytosolic DNA sensor that detects DNA exposed in the cytoplasm either from pathogens or cellular damage. Activated cGAS instigates the signaling cascades to activate type I interferon (IFN) expression, critical for host defense and autoimmune diseases. In addition, cGAS plays a role in senescence, DNA repair, apoptosis, and tumorigenesis. Recently, various post-translational modifications (PTMs) of cGAS have been reported, such as phosphorylation, ubiquitination, acetylation, glutamylation, and sumoylation. These PTMs profoundly affect cGAS functions. Thus, here we review the recent reported PTMs of cGAS and how these PTMs regulate cGAS enzymatic activity, DNA binding, and protein stability, and discuss the potential future directions.

## 1. Introduction

Innate immunity is the front line of host defense to prevent the infection of pathogens. The innate immune system comprises a variety of signaling pathways that are elicited by microbial components. Usually, these microbial components are distinct from those in the host, which marks the invading pathogens. For example, lipopolysaccharides (LPS) in Gram-negative bacteria’s outer membrane is a foreign product to the host and triggers a potent innate immune response. These microbial components are termed pathogen-associated molecular patterns (PAMPs). The PAMPs are detected by a group of sensors called pattern recognition receptors (PRRs). Once engaged, the PRRs instigate a signal cascade to induce the expression of cytokines and chemokines, thereby limiting the infection.

The PRRs have three major subcellular localizations, the plasma membrane, endosomes, and the cytosol. For example, toll-like receptor 4 (TLR4) expresses in the plasma membrane to detect LPS, whereas TLR7 resides in the endosomes to sense viral RNA. The plasma membrane and endosomes provide a physical barrier to block the PRRs to access cellular contents. On the contrary, without a physical barrier, cytosolic PPRs must discriminate self- and non-self-components, such as the nucleic acids exposed in the cytoplasm. It is well known that viral RNAs present two distinct forms, double-stranded RNA (dsRNA) or 5′ triphosphate single-stranded RNA (ssRNA), which are different from the abundant 5′-capped host mRNA in the cytosol. The dsRNA or 5′ triphosphate ssRNA is sensed as the non-self RNA by the cytosolic RNA sensors, retinoic acid-inducible gene I (RIG-I)-like receptors (RLRs), thus inducing type I interferon (IFN) expression.

Like dsRNA, foreign DNA is also a potent PAMP that activates type I IFN and pro-inflammatory cytokines. Many reported DNA sensors detect cytosolic DNA and elicit type I IFN production. However, cyclic GMP–AMP synthase (cGAS), the recently identified cytosolic PRR, is the widely accepted cytosolic DNA sensor that detects pathogenic DNA exposed in the cytoplasm [1]. As a member of the nucleotidyltransferase (NTase) enzyme family, cGAS comprises an unstructured amino-terminal (N-terminal) domain followed by the NTase domain and a carboxyl-terminal (C-terminal) domain. Upon DNA binding, cGAS undergoes conformational changes to form a dimer, which activates cGAS enzymatic activity and produces cyclic GMP–AMP (cGAMP) using ATP and GTP [1,2,3,4,5,6,7,8,9]. cGAMP is a second messenger that binds the endoplasmic reticulum membrane protein, a stimulator of interferon genes (STING), which leads to the multimerization of STING and intracellular trafficking from the endoplasmic reticulum (ER) to the endoplasmic-reticulum–Golgi intermediate compartment (ERGIC) or the Golgi apparatus. On the ERGIC, STING recruits and activates TANK-binding kinase 1 (TBK1). Activated TBK1 phosphorylates interferon regulatory factors (IRFs), which triggers the dimerization and nuclear translocation of IRFs. In the nucleus, IRFs form active transcriptional complexes and activate type I IFN gene expression. Type I IFN further induces a cohort of interferon-stimulated genes to eradicate pathogens (Figure 1).

It has not been reported that cGAS can discriminate pathogenic and host DNA, i.e., the preference for DNA sequence and modification has yet to be found. Excessive host DNA activates the cGAS signaling pathway, leading to aberrant IFN activation and autoimmune diseases, such as systemic lupus erythematosus (SLE). Therefore, cells must keep cGAS silent from host genomic DNA and, paradoxically, at the same time, cells need to keep cGAS agile to foreign DNA. The old paradigm is that host DNA normally is restricted to cellular compartments, such as the nucleus and the mitochondria. As cGAS was first thought to be a solely cytosolic protein, the physical barrier blocks the access of cGAS to host DNA. Later studies further found additional subcellular localizations of cGAS, including predominantly in the nucleus [10], on the plasma membrane [11], mitosis-associated nuclear localization [12,13], or phosphorylation-mediated cytosolic retention [14]. One study showed that phosphoinositide interactions anchor cGAS at the plasma membrane to avoid self-DNA detection and activation [11]. Nonetheless, the nuclear localization of cGAS has been validated by many laboratories independently [10,13,15] and is of particular interest as the nucleus is a DNA-rich environment and nuclear cGAS is tethered to the chromatin. Five recent structural works revealed that cGAS binds to the H2A-H2B dimer of the nucleosome and the binding immobilizes cGAS on the chromatin, so cGAS cannot access the nearby DNA to form an active dimer [16,17,18,19,20]. Although the mechanism of nuclear cGAS self-inhibition is elucidated, understanding the role of cGAS in the nucleus has just begun.

Sensing cytosolic self-DNA via cGAS has been shown to have profound pathological effects. The cytosolic presence of micronuclei, mitochondrial DNA (mtDNA), endogenous retroelements, telomeric DNA, chromatin, or DNA leaked from defective DNA replication, repair, and mitosis has been found to be sensed by cGAS [21]. Sensing these self-DNA highlights the critical role of cGAS in pathological processes, such as senescence and cell death. In senescent cells, chromatin fragments are leaked into the cytoplasm and cause the senescence-associated secretory phenotype (SASP). SASP is featured with high levels of inflammatory cytokines and requires cGAS activation [12,22,23,24]. In addition, cGAS promotes cell death through two mechanisms under different contexts. First, cGAS announces mitotic arrest-induced cell death. During the mitotic arrest, cGAS-dependent IRF3 phosphorylation slowly accumulates without triggering inflammation. Phosphorylated IRF3 stimulates apoptosis through alleviating Bcl-xL-dependent suppression of mitochondrial outer membrane permeabilization [25]. Secondly, cGAS is also required for autophagy-mediated cell death during a replicative crisis. The leaked cytosolic chromatin fragments during the crisis activate the cGAS–STING pathway and promote autophagy [26].

Recent studies found that the role of cGAS is beyond the canonical cGAS–STING pathway and more intricate than expected [27]. cGAS was found to respond to directly regulates DNA repair in a cGAMP-independent manner. cGAS inhibits the double-strand DNA break repair by homologous recombination, which does not require cGAS catalytic activity and activation of the cGAS–STING signaling pathway [14,28]. cGAS is recruited to double-stranded breaks and interacts with the poly (ADP-Ribose) polymerase 1 (PARP1). The cGAS–PARP1 interaction impedes the formation of the PARP1–Timeless complex, and thereby suppresses homologous recombination. Furthermore, knockdown of cGAS suppresses DNA damage and inhibits tumor growth both in vitro and in vivo, suggesting cGAS as a tumor enhancer [14]. Although the oncogene-like role of cGAS is contradictory to the positive role of cGAS in cell death as discussed above, the complexity might be context-dependent like the reported differential roles of the cGAS–STING pathway in cancers [29].

Several co-factors have been reported to regulate cGAS activity. Polyglutamine-binding protein 1 (PQBP1) binds to reverse-transcribed HIV-1 DNA and interacts with cGAS for efficient IRF3 activation [30]. The GTPase-activating protein SH3 domain-binding protein 1 (G3BP1) regulates cGAS activation by enhancing cGAS DNA binding and promoting the formation of large cGAS complexes [31]. Interestingly, ZCCHC3, a CCHC-type zinc-finger protein, also enhance cGAS DNA binding to promote cGAS activity [32]. In addition to the regulations by co-factors, cGAS activity is regulated on transcriptional and translational levels. Firstly, although cGAS is constitutively expressed, its expression can be further induced by IFN [33]. The increased expression of cGAS amplifies innate immune responses. Secondly, cGAS is subject to different post-translational modifications (PTMs), including phosphorylation, ubiquitylation, acetylation, glutamylation, and sumoylation (Table 1). These PTMs play a critical role in regulating cGAS enzymatic activity, DNA binding, protein stability, and subcellular localization, consequently contributing to the versatile functions of cGAS (Figure 2). PTMs are usually mediated by enzymes, which are ideal drug targets. Elucidation of PTMs will help develop novel therapeutics targeting cGAS, boosting host immunity to inhibit pathogen infection or downregulate innate immune response to ameliorate autoimmune disease. Thus, we provide a review below on each PTM and its regulatory mechanism for cGAS.

## 2. Post-Translational Modifications of cGAS

### 2.1. Phosphorylation

Phosphorylation is the reversible PTM at serine (S), threonine (T), and tyrosine (Y) in eukaryotes, which is regulated by kinase and phosphatase. Phosphorylation has been shown to regulate many cellular processes and signaling pathways, including the innate immune signaling pathway. Recent studies showed that phosphorylation also regulates cGAS enzymatic activity and subcellular localization. It has been reported that the S291 site of mouse cGAS (S305, an equivalent site in human cGAS) can be phosphorylated by two kinases, AKT serine/threonine kinase 1 (AKT1) and cyclin-dependent kinase 1 (CDK1) [34,35]. S291 is proximal to the catalytic sites. S291 phosphorylation robustly suppresses cGAS enzymatic activity in asynchronous [34] and mitotic cells [35], although how this phosphorylation alters the cGAS protein conformation to suppress enzymatic activity is not known. AKT1-mediated S291 phosphorylation of cGAS leads to the reduction of cGAMP and IFNβ production and the increase of herpes simplex virus 1 (HSV-1) replication. Conversely, the AKT inhibitor VIII and the S291A mutant of cGAS enhance DNA-induced IFNβ production and inhibit HSV-1 infection [34]. AKT1, also known as protein kinase B (PKB), regulates many processes, such as proliferation, metabolism, growth, and cell survival. It will be interesting to investigate how these processes crosstalk with AKT1-mediated suppression of innate immunity. Additionally, many viruses hijack the PI3K-AKT1 signaling pathway for effective viral replication [50]; however, the underlying molecular mechanism is not well elucidated. Hence, future work will need to investigate whether AKT activation is a viral evasion strategy via phosphorylation of cGAS.

Unlike AKT, CDK1 only phosphorylates the S291 site of cGAS during mitosis, preventing cGAS activation from host genomic DNA [35]. The same study also showed that protein phosphatase 1 (PP1) dephosphorylates the S291 site of cGAS upon mitotic exit, thereby restoring cGAS ability to sense DNA in the cytosol. The cell cycle-dependent phosphorylation provides an explanation of why cGAS is inert when bound to the chromatin during mitosis. Although recent structural works found that the immobilization of cGAS by H2A-H2B dimer blocks the access to DNA, CDK1-mediated phosphorylation adds another layer of regulation. Furthermore, as the S291A mutant cannot be phosphorylated by CDK1 and shows a full enzymatic activity in vitro [34], it will be interesting to investigate whether the S291A mutant is active during mitosis and in the nucleus.

PPP6C, the catalytic subunit of PP6, is also found to dephosphorylates cGAS but at a different site, S420 in mouse (S435 in human) [36]. PPP6C is constitutively associated with cGAS in un-stimulated cells and disassociated upon virus infection. The disassociation of PPP6C results in S420 phosphorylation in the catalytic pocket of cGAS. The phosphorylated S420 is required for cGAS to bind GTP and to generate cGAMP [36]. Unphosphorylated cGAS purified from bacteria can produce cGAMP in vitro, although the S420 phosphorylated one shows higher activity [36], suggesting that this site is fine-tuning GTP binding. Moreover, the kinase for S420 is still in search.

In addition to serine phosphorylation, cGAS is also subject to tyrosine phosphorylation, which controls cGAS cytosolic retention [14]. The study showed that DNA damage induces nuclear translocation of cGAS in a manner that is dependent on importin-α, and the nuclear cGAS suppresses DNA repair [14]. The B lymphocyte kinase (BLK kinase) phosphorylates at the Y215 site of human cGAS, facilitating the cytosolic retention of cGAS [14]. However, other studies did not find evidence of cGAS nuclear import following DNA damage [13,28]. Furthermore, as discussed above, cGAS nuclear localization has been found under normal physiological conditions. Nonetheless, it is still possible that BLK-mediated Y215 phosphorylation induces the export of preexisted nuclear cGAS to the cytoplasm, thereby blocking nuclear cGAS-mediated suppression of DNA repair.

### 2.2. Acetylation

Lysine acetylation is another reversible PTM that plays a crucial role in the regulation of chromatin structure, protein function, protein stability, subcellular localization, and gene expression [51]. Although lysine acetylation was first discovered in histones, subsequent studies further found the lysine acetylation of non-histone proteins. Acetylation of histones or non-histone proteins is mainly reversibly regulated by lysine acetyltransferase and lysine deacetylase. Recent studies found that cGAS activity was regulated by acetylation [37,38]. Interestingly, both acetylation and deacetylation activate cGAS, although via different sites [37,38]. It was first reported that deacetylation is required for cGAS activation [38]. The study showed that cGAS acetylation on either K384, K394, or K414 contributes to keeping cGAS inactive. Upon DNA stimulation, cGAS is deacetylated by the histone deacetylase 3 (HDAC3). The mechanism of how deacetylation activates cGAS is unknown. However, these three sites are located in the regions for DNA binding, catalytic activity, and dimerization, so adding an acetyl group onto these sites might directly impact cGAS activation. More strikingly, the study also showed that aspirin can directly acetylate cGAS and efficiently inhibit cGAS-mediated immune responses [38]. As a non-steroidal anti-inflammatory drug (NSAID), aspirin plays a much broader role by acetylation of various other substrates. Furthermore, the cognate acetyltransferase for the three lysines in cGAS is unknown. A later study showed that the lysine acetyltransferase 5 (KAT5) is an acetyltransferase for cGAS; however, the acetylation promotes cGAS DNA binding activity and activation [37]. The discrepancy of these two studies is due to the fact that KAT5 acetylates cGAS at different sites, including K47, K56, K62, and K83 in the N-terminal domain [37]. These data demonstrate the complexity of PTM and the same PTM on different sites might cause opposite consequences.

### 2.3. Glutamylation

The glutamylation is an ATP-dependent process that adds glutamate chains onto the conserved glutamate residues in the target proteins [52,53], which is catalyzed by tubulin tyrosine ligase (TTL) and tubulin tyrosine ligase-like (TTLL) enzymes. The glutamates can be removed by a family of cytosolic carboxypeptidases (CCPs) [53]. The process of glutamylation is evolutionarily conserved from protists to mammals, and the most prominent substrate is tubulin, the microtubule building block [54]. Glutamylation is generally recognized as a modification of tubulin; however, it is not restricted to tubulin. A recent study found that glutamylation and deglutamylation of cGAS modulates immune responses to infection with DNA viruses, evidenced by the fact that *Ccp5*^−/−^ or *Ccp6*^−/−^ mice are more susceptible to DNA virus infection [39]. The study further found that the E272 and E302 sites of mouse cGAS are glutamylated by TTLL6 and TTLL4, respectively. TTLL6-mediated polyglutamylation at E272 dampens the DNA binding activity of cGAS, whereas TTLL4-mediated monoglutamylation at E302 blocks cGAS enzymatic activity [39]. Conversely, CCP6 removes the polyglutamylation of cGAS, whereas CCP5 hydrolyzes the monoglutamylation of cGAS, which together leads to the activation of cGAS [39]. The study also found that TTLL4 and TTLL6 protein decreased substantially after infection with HSV. It suggests a regulatory mechanism for these enzymes; however, how these enzymes are activated under the context of viral infection and the dynamics of the enzymatic activities are not known.

### 2.4. Ubiquitination

Ubiquitin is a highly conserved, ubiquitously expressed small protein. The ubiquitination process is a reversible covalent conjugation of ubiquitin to substrate by a stepwise enzymatic reaction involving ubiquitin-activating enzyme (E1), ubiquitin-conjugating enzyme (E2), and ubiquitin ligase (E3). Ubiquitin is first activated by the E1 in an ATP-dependent reaction by linking the C-terminal carboxyl group of ubiquitin to the sulfhydryl group of the E1 through a thioester bond. Next, the activated ubiquitin is transferred to an E2, forming an E2-Ub thioester. Lastly, the E2 works with an E3 and conjugates ubiquitin to the substrate protein via an isopeptide bond between the C-terminal glycine of ubiquitin and the lysine of the substrate protein. The E3 ubiquitin ligase determines the substrate specificity. In addition, the C-terminus of ubiquitin can be conjugated to one of the seven lysines (K6, K11, K27, K29, K33, K48, K63) on another ubiquitin, forming a polyubiquitin chain of different linkages. The linkage of polyubiquitin affects the fate of the substrate and plays a critical role in various signaling pathways [55]. For example, K48-linked ubiquitination is usually involved in directing proteins for proteasome-dependent degradation. In contrast, the K63-linked ubiquitination involves non-proteolytic processes, such as subcellular localization, trafficking, protein stability, and signalosome activation.

It has been reported that cGAS is polyubiquitinated via different linkages by several ubiquitin E3 ligases. It was first reported that the ER ubiquitin ligase RNF185 interacts with cGAS during HSV-1 infection. Then, RNF185 specifically catalyzes the K27-linked polyubiquitination of cGAS at two sites, K173 and K184, promoting cGAS enzymatic activity [40]. Another ubiquitin E3 ligase, TRIM56, also positively regulates cGAS activity [41]. The study showed that TRIM56 mediates monoubiquitination of cGAS at K335, resulting in a marked increase in its dimerization, DNA-binding activity, and cGAMP production, consequently promote the antiviral responses [41]. Similar to TRIM56, another TRIM E3 ligase, TRIM41, also binds cGAS and mediates monoubiquitination of cGAS to promote its activation [42]. In addition, TRIM14, a TRIM protein without the N-terminal RING domain, promotes cGAS activation [43]. Although TRIM14 does not possess E3 ligase activity, it acts as an adaptor to recruit USP14, a deubiquitinating enzyme. USP14 cleaves the K48-linked polyubiquitin chains of cGAS at K414, thereby inhibiting p62-mediated autophagic degradation of cGAS to enhance the activation of type I interferon signaling [43]. Interestingly, the deubiquitinase, USP27X, also interacts with cGAS and cleaves K48-linked polyubiquitination chains from cGAS, leading to cGAS stabilization [44]. However, the E3 ligase(s) responsible for cGAS K48-linked polyubiquitination is unknown. Searching the E3 ligase(s) for cGAS K48-linked polyubiquitination will further elucidate the regulatory mechanism controlling cGAS protein stability.

### 2.5. SUMOylation

The small ubiquitin-related modifier (SUMO) proteins are a group of ubiquitin-like proteins, including SUMO1-4. Similar to the process of ubiquitination, SUMO is covalently conjugated to a lysine residue in a target protein by a stepwise enzymatic reaction [56]. SUMOylation can affect protein function in various ways, such as protein stability and subcellular localization [57]. Recently, it was reported that the ubiquitin ligase TRIM38 targets mouse cGAS for sumoylation at K217 and K464 [45]. Interestingly, the same study also showed that TRIM38 also SUMOylates STING, the downstream effector of cGAS. Furthermore, TRIM38 mediates cGAS SUMOylation in uninfected cells and during the early phase of viral infection, which promotes cGAS protein stability by preventing cGAS from K48-linked polyubiquitination and degradation [45]. The study also found that the SUMO1 specific peptidase 2 (SENP2) deSUMOylates cGAS and STING at the late phase of viral infection or DNA stimulation, which conditions them for subsequent degradation by the ubiquitin-proteasomal or chaperone-mediated autophagy [45]. How SENP2 selectively functions in the late phase of viral infection is unknown. Nonetheless, the data suggest that deSUMOylation negatively regulates cGAS activity. By contrast, a later study found that deSUMOylation activates cGAS activity by a different SUMO specific peptidase, SENP7 [46]. They found that cGAS is SUMOylated at K335, K372, and K382, although the responsible E3 ligase(s) is yet to be discovered. SUMOylation of these sites leads to the suppression of DNA binding, oligomerization, and nucleotidyltransferase activities of cGAS. SENP7 reverses this inhibition via catalyzing the cGAS deSUMOylation. The discrepancy of these studies may lie on the SUMOylation type and sites; therefore, future investigations need to determine how these SUMOylations coordinate to regulate host innate immune responses.

### 2.6. Caspase-Mediated Cleavage

Apoptosis is a programmed cell death with dramatic changes, such as nuclear blebbing, nuclear fragmentation, and mitochondrial damage. The damage of the nucleus and the mitochondria during apoptosis leads to leakage of nuclear DNA and mtDNA, which might activate cGAS and induce type I IFN production. However, the activated caspase cascade responsible for facilitating cell death also prevents dying cells from triggering a host immune response [47]. White et al. showed that Bak and Bax trigger the release of mitochondrial DNA during apoptosis, which activates the cGAS signaling pathway and elicits type I IFN expression [47]. Meanwhile, the activated caspases render mtDNA-induced innate immunity silent, as evidenced by the observations that pharmacological caspase inhibition or genetic deletion of caspase-9, caspase-3/7, or Apaf-1 causes dying cells to produce type I IFN [47]. A later study further showed that the activated caspase-3 cleaves cGAS to prevent cytokine overproduction in virus-induced apoptosis [49]. The activated caspase-3 also cleaves MAVS, a downstream adaptor of the RNA sensor, RIG-I, and the transcriptional factor IRF3 to keep apoptosis immunologically silent [49]. The study further showed that caspase-3 is required in human and mouse cells, whereas caspase-7 is involved only in mouse cells to inactivate cGAS, suggesting a species-dependent mechanism. The caspase-3 cleaves human cGAS at D319, a conserved site among vertebrates, located on the central β sheet inside the cGAS catalytic pocket. Thus, the cleavage leads to loss of enzymatic activity of cGAS.

Interestingly, the same group on the capsase-3 study also reported a caspase-1-mediated cleavage of cGAS [48]. Unlike caspase-3 or 9, caspase-1 is not involved in apoptosis. Instead, caspase-1 is critical for inflammasome activation by proteolytically cleaving and activating the inactive precursor of interleukin-1 (IL-1). The study showed that caspase-1 cleaves human cGAS at D140 and D157 during inflammasome activation, resulting in reduced cGAMP production and cytokine expression. Consistently, caspase-1-deficient mice show an enhanced resistance to infection by DNA but not RNA viruses [48]. The study further showed that cGAS is also cleaved by caspase-4, caspase-5, and caspase-11 during noncanonical inflammasome activation triggered by LPS. The study concludes that inflammasome activation dampens cGAS-dependent signaling, suggesting a crosstalk between the intracellular DNA-sensing pathway and the inflammasome pathway. It is well known that cytosolic DNA is sensed by cGAS and the absent in melanoma 2 (AIM2) to activate type I IFN and inflammasome, respectively. The imbalanced output of these two pathways will cause pathological consequences. Future work will need to investigate the effects of temporal and spatial DNA sensing on these two pathways and how the interactions between these two pathways contribute to the distinct pathological outcomes.

## 3. Conclusions and Perspectives

The recent advances of cGAS, especially the discovery of nuclear localization, raises many open questions. First, what is the regulatory mechanism for the nuclear-cytoplasmic shuttling of cGAS? How is nuclear cGAS released from the chromatin? Secondly, it has been shown that cGAS senses HIV-2 viral DNA in the nucleus with the assistance of the non-POU domain-containing octamer binding protein (NONO) [58]. Since most DNA viruses replicate in the nucleus, is nuclear cGAS also able to sense their DNA in the nucleus? Furthermore, does the nuclear stress or DNA damage caused by the replication of a DNA virus lead to the release of cGAS from the chromatin? Thirdly, can the nuclear cGAS act as a sensor for the danger signal from genomic DNA damage in situ? Fourthly, does cGAS alter histone PTMs and affect transcription or even epigenetics? Lastly, how do PTMs regulate cGAS in these aspects? Current PTM studies focus on cytosolic cGAS. It is not clear whether these reported PTMs also regulate nuclear cGAS. Are there any PTMs in nuclear cGAS? It will be interesting to discover the PTMs of nuclear cGAS and investigate their biological activities in the future.

Most PTM sites are found on the NTase and C-terminal domain of cGAS. Although the N-terminal domain deletion mutant of cGAS can be fully activated in vitro or by overexpression in cells [1], the activation of dendritic cells mediated by cGAS signaling was lost upon the deletion of the N-terminal domain [13]. The N terminus of cGAS is required for DNA-induced liquid phase condensation of cGAS and activation of innate immune signaling [59]. A recent study showed that the N-terminal domain determines the nuclear localization of cGAS [13]. The challenge of a study of the N-terminal domain of cGAS is the lack of high conservation among species. Nonetheless, it will be interesting to investigate whether and how the PTMs in the N terminus regulate cGAS activity and subcellular localization.

The emerging roles of cGAS in senescence and apoptosis suggest that cGAS is involved in tumorigenesis and metastasis. Whether and how PTMs of cGAS affect cancer cell death are not clear. It is notable that cGAS knockout mice are healthy and normally developed in a barrier facility [60]. Furthermore, cytosolic DNA sensing through cGAS and STING is inactivated due to gene mutations in wild pangolins [61]. Thus, future works will be needed to use genetic tools to establish mouse cancer models in cGAS deficiency or PTM deficiency backgrounds. It would also be interesting to mine the GWAS data to seek the correlations between potential cGAS mutations, including mutations in PTM sites and human diseases. It is predicted that more PTMs will be discovered under versatile contexts using powerful omics tools, such as proteomics. For example, a recent proteomic study found five phosphorylation and six acetylation sites that have not been previously documented [62]. The role of new PTMs will help elucidate the regulatory mechanisms for cGAS, and the PTM sites will be ideal drug targets for therapeutics in autoimmune diseases and cancers.

## Figures and Tables

**Figure 1 ijms-21-07842-f001:**
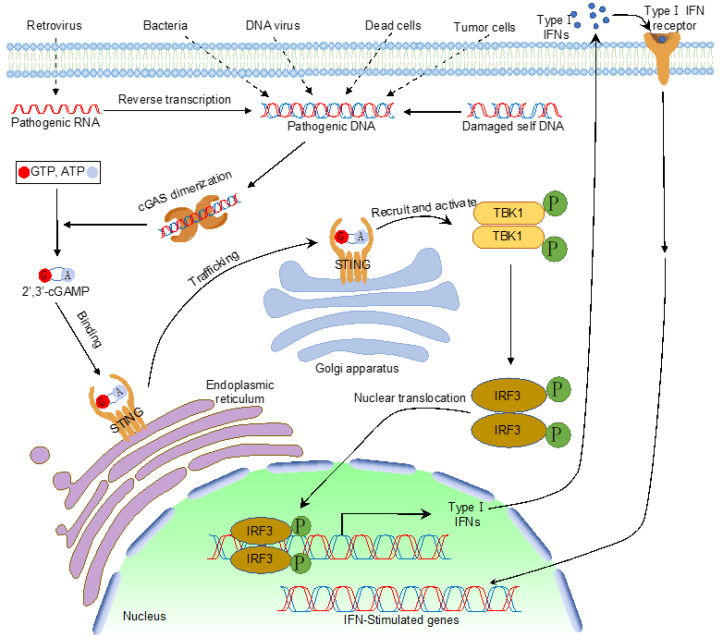
The cytosolic DNA sensing pathway mediated by cyclic GMP–AMP synthase (cGAS).

**Figure 2 ijms-21-07842-f002:**
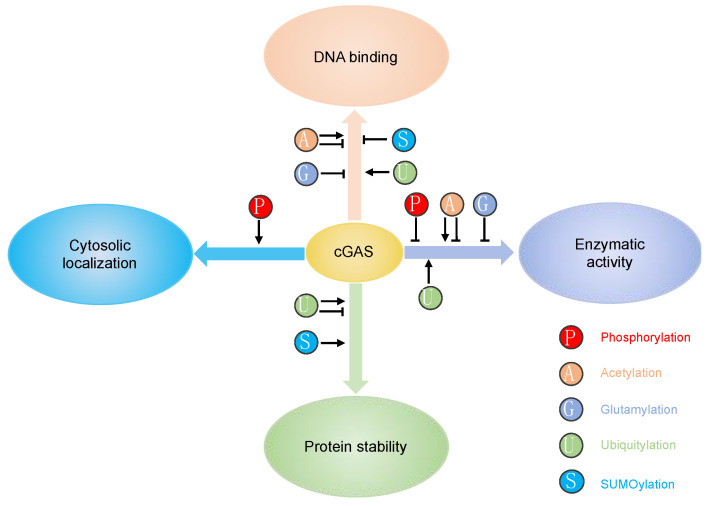
PTMs play a critical role in regulating cGAS enzymatic activity, DNA binding, protein stability and subcellular localization.

**Table 1 ijms-21-07842-t001:** The cGAS post-translational modification (PTM) sites and their relative roles.

PTM Site (h: Human cGAS; m: Mouse cGAS)	PTM	Enzyme	Function	Reference
S291 (m)	Phosphorylation	AKT	Suppresses cGAS enzymatic activity	[34]
S291 (m)	Phosphorylation	CDK1	Suppresses cGAS enzymatic activity during mitosis	[35]
Y215 (h)	Phosphorylation	BLK	Facilitates the cytosolic retention of cGAS	[14]
S291 (m)	Dephosphorylation	PP1	Restores cGAS activity in the cytoplasm upon mitotic exit	[35]
S420 (m)	Dephosphorylation	PPP6C	Prevents cGAS from binding to GTP and inhibit cGAS activity	[36]
K47 (h), K56 (h), K62 (h), and K83(h)	Acetylation	KAT5	Promotes cGAS DNA binding activity and activation	[37]
K384 (h), K394 (h), and K414 (h)	Deacetylation	HDAC3	Required for cGAS activation	[38]
E272 (m)	Polyglutamylation	TTLL6	Dampens the DNA binding activity of cGAS	[39]
E302 (m)	Monoglutamylation	TTLL4	Blocks cGAS enzymatic activity	[39]
E302 (m)	Deglutamylation	CCP5	Removes the monolutamylation of cGAS	[39]
E272 (m)	Deglutamylation	CCP6	Removes the polyglutamylation of cGAS	[39]
K173 (m) and K184 (m)	Polyubiquitination	RNF185	Medaites K27-linked polyubiquitination and promotes enzymatic activity of cGAS	[40]
K335 (m)	Monoubiquitination	TRIM56	Promotes cGAS dimerization and DNA-binding activity	[41]
Unknown	Monoubiquitination	TRIM41	Promotes cGAS activation	[42]
K414 (h)	Deubiquitination	USP14	Cleaves the K48-linked polyubiquitination of cGAS and prevents cGAS degradation	[43]
Unknown	Deubiquitination	USP27X	Cleaves the K48-linked polyubiquitination of cGAS and prevents cGAS degradation	[44]
K217 (m) and K464 (m)	SUMOylation	TRIM38	Prevents cGAS from K48-linked polyubiquitination and degradation	[45]
K217 (m) and K464 (m)	DeSUMOylation	SENP2	Promotes cGAS degradation	[45]
K335 (m), K372 (m), and K382 (m)	DeSUMOylation	SENP7	Promotes cGAS dimerization and DNA-binding activity	[46]
Unknown	Cleavage	Caspase 9	Suppresses mtDNA-induced type I IFN production	[47]
Unknown	Cleavage	Caspase 3/7	Suppresses mtDNA-induced type I IFN production	[47]
D140 (h) and D157 (h)	Cleavage	Caspase 1	Inhibits cGAS activity	[48]
D319 (h)	Cleavage	Caspase 3	Inhibits cGAS activity	[49]

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
