# Peer review of "Role of Post-Translational Modifications of cGAS in Innate Immunity"

_ijms, 2020, doi:10.3390/ijms21217842_

Round 1

Reviewer 1 Report

A comprehensive review describing the cGAS sensor and its regulation by post-translational modifications.

In the introduction to cGAS sensing, I suggest mentioning that the pathway is not required in probably very rare cases in mammals. It has been described as missing in pangolins (PMID:32533513).

Minor English corrections required, please check e.g lines 22, 36, 73, 94, 103, 164, 178, 304, 309.

Author Response

We thank the reviewer for the constructive suggestions. In the revision, we added the reference and make corrections on the indicated lines. All changes are highlighted in the text. 

Reviewer 2 Report

The Review write by Wu and Li in my opinion is well organized and describes in a comprehensive way the role of Post-Translational Modifications of cGAS in Innate Immunity.

I think it can be accepted in this form

Author Response

We appreciate the reviewer for the positive comments.

Reviewer 3 Report

This review provides a comprehensive survey of the recent knowledge on the role of cGAS in antimicrobial innate immunity. Furthermore, it discusses extensively the known post-translational modifications and its consequences in the protein function. I think this review is fit for IJMS and I strongly recommend its acceptation. A short abbreviation list could help the better understanding by non-experts, but interested readers.

Author Response

We thank the reviewer for the positive comments. A short abbreviation list is added as Supplementary Table 1 in the revision.